# An Improved YOLOv8 OBB Model for Ship Detection through Stable Diffusion Data Augmentation

**DOI:** 10.3390/s24175850

**Published:** 2024-09-09

**Authors:** Sang Feng, Yi Huang, Ning Zhang

**Affiliations:** School of Electromechanical Engineering, Guangdong University of Technology, Guangzhou 510006, China; 18076696122@163.com (Y.H.); 15521108851@163.com (N.Z.)

**Keywords:** ship detection, YOLOv8, stable diffusion, BiFPN structure, EMA module, multi-scale detection

## Abstract

Unmanned aerial vehicles (UAVs) with cameras offer extensive monitoring capabilities and exceptional maneuverability, making them ideal for real-time ship detection and effective ship management. However, ship detection by camera-equipped UAVs faces challenges when it comes to multi-viewpoints, multi-scales, environmental variability, and dataset scarcity. To overcome these challenges, we proposed a data augmentation method based on stable diffusion to generate new images for expanding the dataset. Additionally, we improve the YOLOv8n OBB model by incorporating the BiFPN structure and EMA module, enhancing its ability to detect multi-viewpoint and multi-scale ship instances. Through multiple comparative experiments, we evaluated the effectiveness of our proposed data augmentation method and the improved model. The results indicated that our proposed data augmentation method is effective for low-volume datasets with complex object features. The YOLOv8n-BiFPN-EMA OBB model we proposed performed well in detecting multi-viewpoint and multi-scale ship instances, achieving the mAP (@0.5) of 92.3%, the mAP (@0.5:0.95) of 77.5%, a reduction of 0.8 million in model parameters, and a detection speed that satisfies real-time ship detection requirements.

## 1. Introduction

Inland waterway transportation promotes economic activities between inland regions and coastal areas. The Pearl River is an essential and efficient inland waterway connecting the Guangdong-Hong Kong-Macao Greater Bay Area with the southwestern regions of China. As the region’s economy develops, ship traffic in the Pearl River waterway continues to increase rapidly [1,2]. The substantial ship traffic poses a significant challenge to the management of the Pearl River waterway. The effective detection of ships is crucial to the management and monitoring of inland waterway transportation. Shore-based cameras are frequently used to monitor ships through optical images [3]. Compared with shore-based cameras, camera-equipped Unmanned Aerial Vehicles (UAVs) have the advantages of more significant monitoring coverage and more excellent maneuverability. With the development of UAV technology, UAVs are increasingly used in maritime and inland waterway transportation [4,5,6,7]. 

Object detection, a crucial task in computer vision, is applied to ascertain the classifications and positions of various objects. Traditional object detection methods are based on handcrafted features like LBP, HAAR, HOG, etc. [8,9,10,11]. With the rapid development of deep learning techniques, feature extraction techniques for convolutional neural networks (CNNs) are being widely used in object detection methods. Compared to traditional object detection methods, CNN-based object detection methods achieve faster detection speeds and higher accuracy. Many CNN-based object detection methods have been proposed in recent years, such as R-CNN, Fast R-CNN, YOLO, etc. Region with CNN feature (R-CNN) method is the first time that convolutional neural networks are applied in object detection tasks [12]. With outstanding feature extraction capability, CNNs significantly enhance the effectiveness of object detection. The Fast R-CNN method is proposed based on the R-CNN method and previous work to efficiently classify object proposals using deep convolutional networks, achieving faster speed and higher accuracy in object detection [13]. The YOLO method is famous for its quick detection speed and high accuracy and is widely used in real-time detection tasks [14]. After years of development, the YOLO method has evolved into many versions.

Ship detection is a typical object detection task widely used in ship navigation, waterway management, and maritime rescue [15,16]. Ship detection by camera-equipped UAVs significantly enhances the efficiency of ship management in waterways. To address the challenges of ship detection in UAV-captured views and limited computing resources of UAV platforms, Wang et al. [17] proposed a YOLOv7-DyGSConv model for ship detection in real-time videos captured by UAVs, which enhances detection accuracy and speed through an attention mechanism and GSConv integration. By incorporating ODConv and ConvNeXt blocks into the YOLOv5 backbone, Cheng et al. [18] proposed a YOLOv5-ODConvNeXt model to accelerate ship detection speed and boost detection accuracy. Aiming at decreased accuracy in detecting small-sized and densely packed ships, Li et al. [19] proposed an improved YOLOv5 model to detect ships accurately under UAV vision and combined with deepsort to realize ship tracking. They added a detection layer to better utilize detailed shallow features and introduced coordinate attention to enhance the focus on critical feature information.

To improve the management efficiency of the Pearl River waterway, this paper conducts a study on ship detection by camera-equipped UAVs. In this study, ship detection is more challenging due to the following issues:(1)Multi-viewpoint of aerial images. As UAVs take images in the air, ships may appear from different viewpoints, such as front, side, rear, etc. [20].(2)Multiple scales of ship instances. The different distances between UAVs and ship instances lead to ship instances having numerous scales, and various types of ships also have multiple scales.(3)Environment variability. Natural illumination may change with the weather. Due to some interference factors, the background of aerial images is often cluttered. For instance, white caps caused by waves may change the background [21].(4)Dataset scarcity. Due to different hydrological conditions, the features of ships vary from one water area to another. In order to ensure the effectiveness of this work in the Pearl River Waterway, it is necessary to collect the images of ships in the Pearl River Waterway and then make a dataset of them.

Most CNN-based object detection algorithms use horizontal boxes as bounding boxes, but horizontal boxes are not suitable for ship detection by UAVs. Due to the multiple viewpoints of aerial images and the elongated shape of ships, horizontal boxes would contain too much background. Oriented object detection goes a step further than object detection and introduces an extra angle to locate objects more accurately in an image. Oriented object detection algorithms use oriented bounding boxes. Figure 1 shows a horizontal bounding box and an oriented bounding box enclosing the same ship instance. Therefore, oriented object detection algorithms are suitable for ship detection by UAVs. By modifying the Faster R-CNN algorithm, R2CNN is the first algorithm to support oriented boxes [22]. Since then, several oriented object detection algorithms have been proposed based on multiple object detection algorithms, such as Oriented R-CNN [23], Rotated Faster R-CNN [24], Improved YOLO [25], etc. The YOLOv8 OBB algorithm is an oriented bounding box (OBB) object detection algorithm developed by Ultralytics and is the first official version of YOLO algorithms to support oriented bounding boxes. Considering the scalability and excellent performance of the YOLOv8 algorithm, this paper is based on the YOLOv8n OBB algorithm.

The structure of the YOLOv8 network consists of the backbone, neck, and head. Many methods exist to improve the YOLOv8 network’s performance on different tasks. The attention mechanism is an essential technique in deep learning, widely used in various tasks such as natural language processing and computer vision. The attention mechanism mimics the human visual attention system and assigns weights to different parts of the input data to enable the CNNs to process important information more effectively. Adding attention modules is an important method to improve the CNNs, for example, adding the SE module [26], SA module [27], CBAM [28], etc. In addition, improving feature pyramid networks is an excellent method to enhance the YOLOv8 network, as feature pyramid networks help detect multi-scale objects. Based on a lightweight residual feature pyramid network (RE-FPN) structure, Li et al. [29] enhanced the YOLOv8 object detection algorithm and augmented accuracy in detecting small objects. In order to solve the problem of small object detection in remote sensing satellite images, Elhamied et al. [30] employed a distinctive attention-guided bidirectional feature pyramid network to improve the YOLOv8s model and the detection accuracy of small objects.

The quantity and quality of data are critical factors that affect the training of CNNs. Data augmentation methods are often employed when the quantity and quality of raw data are unsatisfactory. Traditional data augmentation for increasing the quality and quantity of raw data includes flipping, rotation, scaling, cropping, panning, and adding noise [31]. Generative Adversarial Networks (GANs) are image generation models based on CNNs, which are capable of generating high-quality images. A GAN consists of a discriminator network and a generator network. By training the discriminator and generator networks separately with the original images, the GAN can generate high-quality images similar to the original images [32]. Over a period of time, many GANs have been proposed, such as Anomaly-GAN [33], CGAN [34], CovidGAN [35], etc. Compared with traditional data augmentation methods, the GAN generates more realistic images that can be used to expand the training dataset and improve the object detection model performance. However, GANs have limitations and drawbacks, such as training instability, high computational resource requirements, long training times, and difficulty tuning generation quality. Recently, based on Latent Diffusion Models (LDMs) [36], stable diffusion is proposed as a deep learning model that can efficiently generate images. Compared with GANs, the stable diffusion model can generate higher-quality images and is convenient for data augmentation. Furthermore, the stable diffusion model requires fewer computational resources and is convenient for tuning generation quality. The stable diffusion model rapidly developed and is widely applied in image generation [37,38,39,40].

Therefore, for real-time ship detection by camera-equipped UAVs, we propose an improved method for data augmentation based on a stable diffusion model and present an improved detection model based on the YOLOv8 OBB model. Inspired by the concept of a Bidirectional Feature Pyramid Network (BiFPN), we improve the neck part of the YOLOv8 model and add Efficient Multi-Scale Attention (EMA) attention modules. The main contributions of this paper are summarized as follows:(1)We use camera-equipped UAVs to capture multi-viewpoint images of ships in the Pearl River waterway and apply our proposed data augmentation method for data augment.(2)Based on the BiFPN structure and EMA module, we integrate the P2 feature layer to improve the neck structure of YOLOv8 and propose the YOLOv8n-BiFPN-EMA OBB model.(3)Through multiple comparative experiments, we validate the effectiveness of our proposed data augmentation method and the efficiency of the YOLOv8n-BiFPN-EMA OBB model, leading to significant conclusions.

The rest of this paper is organized as follows: Section 2 introduces the methods for dataset capture, data augmentation, and model improvements. The experimental evaluation metrics and comparative experiments for the data augmentation method and each model are presented in Section 3. The conclusions drawn from the experimental results are presented in Section 4.

## 2. Materials and Methods

### 2.1. Data Capture

After obtaining permission from the relevant authorities, we systematically capture a dataset of ship images using a camera-equipped UAV deployed in a specific section of the Pearl River waterway. To ensure a comprehensive capture of ship images, we take samples of different ships from multiple viewpoints and distances. The original images have a resolution of 4000 × 2250 pixels. After filtering the images, 633 images are obtained. Out of 633 images, 441 are randomly selected for the training set, 68 for the validation set, and 124 for the test set. According to ship functions, the ships in the images are categorized into five classes: bulk carrier, container ship, oil tanker, passenger ships, and others. Figure 2 shows each of the five classes of ship instances. After annotating the ships in the images, the dataset is created.

### 2.2. Data Augmentation

Limited by the short capture time of the images and the number of ships in the waterway, the images and the ship instances of the raw dataset do not satisfy the training requirements of the YOLOv8 model. The purpose of using the stable diffusion model to generate fictional images is as follows: (1) expand hard-to-sample, remedy low-volume instances, (2) reduce the dataset acquisition cycle time, and (3) balance the amount of data per class to improve the generalization ability of the model. A data augmentation method based on the stable diffusion model is described below.

Given an image $x\;\in \;R^{H\times W\times C}$ in RGB space, the encoder *E* encodes x into a latent representation z=E(x), and the decoder *D* reconstructs the image from the latent space, giving $\bar{x}=D(z)=D(E(x))$, where $z\;\in \;R^{h\times w\times c}$ [36].

With the trained perceptual compression models consisting of the encoder *E* and the decoder *D*, the image can be converted between pixel and latent space in which high-frequency, imperceptible details are abstracted. Compared to the high-dimensional pixel space, the latent space is more suitable for likelihood-based generative models, as they can now focus on the crucial, semantic bits of the data and train in a lower dimensional, computationally much more efficient space.

After the image is converted into the latent space, diffusion processes typically occur in the latent space. The diffusion process is an additive noise process that can be described as
(1)$$q\left(z_{T}|z_{T-1}\right)=N\left(z_{T};\sqrt{1-\beta_{T}}z_{T-1},\beta_{T}I\right),$$
where $z_{T}$ is the latent representation at step T. $z_{T-1}$ is the latent representation at the previous step, and N represents the normal distribution (Gaussian distribution). $\sqrt{1-\beta_{T}}z_{T-1}$ represents the mean of the distribution, which is a scaled version of $z_{T-1}$ where the scaling factor $\sqrt{1-\beta_{T}}$ indicates how much noise is added. $\beta_{T}I$ is the covariance matrix of the noise, where $\beta_{T}$ controls the noise strength and I is the identity matrix.

The reverse diffusion process is the denoising process that denoises the noisy latent representation step by step. The deep learning model U-Net takes the latent noisy image and conditioning as input and is used to predict the noise in latent space. The denoising process is described as
(2)$$p(z_{T-1}|z_{T})=N\left(z_{T-1};\mu_{0}(z_{T},T),\sum_{0}(z_{T},T)\right),$$
where $\mu_{0}(z_{T},T)$ and $\sum_{0}(z_{T},T)$ are the mean and covariance parameterized by a time-conditional U-Net [41].

The objective of training the U-Net model is to make the predicted noise as close as possible to the true noise. The objective of the loss function is to minimize the difference between the generated and true samples, thereby optimizing the model parameters such that the generated sample is as close as possible to the true sample. The loss function is described as
(3)$$L_{LDM}=E_{T,z_{0},\epsilon }\left[∥\epsilon -\epsilon_{0}(z_{T},T){∥}^{2}\right],$$
where ϵ is the true noise added during the diffusion process. $\epsilon_{0}(z_{T},T)$ is the noise predicted by the U-Net model.

After T rounds of the denoising process, the latent representation of the generated image is obtained. The generated image, which is similar to the original image, is obtained by the decoder D reconstructing it from the latent space.

Figure 3 shows the structure of the stable diffusion model. Conditioning refers to external input information, including text prompts, images, semantic maps, etc. This information helps the model generate images that meet specific descriptions or requirements.

Based on the stable diffusion model, we propose an improved method for data augmentation. Figure 4 shows the improved data augmentation method. First, the input image is enhanced with the Contrast Limited Adaptive Histogram Equalization (CLAHE) method. CLAHE [42] is an image enhancement method that effectively improves the detail and contrast of an image while avoiding the noise and artifacts that may be introduced by global histogram equalization. Then, the diffusion model generates several images based on the input images. After that, the generated images undergo random transformations, such as HSV adjustment, Gaussian noise, motion blur, random brightness and contrast adjustments, etc. The large number of generated images results in a significant annotation workload. Therefore, we employ the YOLOv8 OBB model, which has been pre-trained on the raw dataset, to automatically annotate the generated images. An augmented dataset is acquired after manual inspection and correction of the annotations.

### 2.3. Model Framework

The role of the neck in the YOLOv8 model is to merge feature maps from different stages of the backbone to capture information at multiple scales. To better address the challenge of multi-scale ship instances, we improve the neck of the YOLOv8 model based on the structure of BiFPN. Additionally, the improved model merges the shallow feature map P2 and incorporates an EMA module in the neck structure. Figure 5 shows the architecture of the improved YOLOv8 OBB model.

#### 2.3.1. Improved BiFPN

In this study, different distances between UAVs and ships lead to ship instances having multiple scales, and different sizes of ships also have multiple scales. Therefore, enhancing the detection performance for multi-scale objects is a crucial optimization direction for the improved model.

The YOLOv8 model employs a Path Aggregation Network (PANet) as the neck structure. PANet enhances the entire feature hierarchy with accurate localization signals in lower layers by bottom-up path augmentation, which shortens the information path between lower layers and the topmost feature [43]. Figure 6a shows the structure of PANet. Due to the complexity of path augmentation and global context fusion in PANet, substantial computational overhead is incurred.

BiFPN employs a bidirectional feature fusion mechanism, ensuring that information across different layers is effectively integrated. BiFPN considers the varying contributions of different features to object detection and optimizes feature map fusion through weighted feature fusion [44]. Additionally, BiFPN removes nodes with only one input and introduces skip connections between the input and output nodes at the same scale, simplifying the network while introducing more features. In Figure 6b, the structure of PANet is shown. Inspired by the structure of BiFPN, we propose an improved BiFPN as the structure of the YOLOv8 neck to enhance the feature fusion capability and improve the ship detection performance. Figure 6c shows the structure of improved BiFPN. Compared to the original YOLOv8 model, which uses the feature maps P3, P4, and P5 as inputs to the neck, we additionally use the feature map P2 as an input. The feature map P2 contains more low-level features, which can retain more helpful information and enhance the network’s sensitivity to details, thereby improving the detection performance of small objects. New feature maps are formed by performing a top-down and bottom-up weighted fusion of the feature maps to facilitate further detection.

#### 2.3.2. EMA Module

In the improved YOLOv8 OBB model, we integrate the EMA modules at the end of the neck to further improve multi-scale object detection performance. In Figure 7, the structure of the EMA module is shown.

Based on the parallel substructures of the CA module [45], a 3 × 3 kernel is innovatively placed in parallel with the 1 × 1 branch to enable fast responses and aggregate multi-scale spatial structural information. Accordingly, the EMA module utilizes three parallel routes to extract attention weight descriptors from the grouped feature maps. Two parallel routes are in the 1 × 1 branch, and the third route is in the 3 × 3 branch. Specifically, two 1D global average pooling operations encode the channel information along two spatial directions in the 1 × 1 branch. At the same time, a single 3 × 3 kernel is used in the 3 × 3 branch to capture multi-scale feature representations [46].

For any given input feature map $X\in R^{C\times H\times W}$, the EMA module divides X into G sub-features for learning different semantics. The groups-style is defined as $X=[X_{0},X_{i},\ldots ,X_{G-1}],{\;X}_{i}\in R^{C//G\times H\times W}$. Without losing generality, let G≪C and the image processing of the interest region in each subsample is enhanced by the learned attention weight descriptors.

In addition, the EMA module employs a cross-spatial information aggregation method across different spatial dimensions to achieve richer feature fusion. The method involves introducing two different data tensors where one is the output of the 1 × 1 branch, and the other is the output of the 3 × 3 branch. The global spatial information in the outputs of the 1 × 1 branch is encoded using 2D global average pooling. The 2D global pooling operation is formulated as
(4)$$z_{c}=\frac{1}{H\times W}\sum_{j}^{H}\sum_{i}^{W}x_{c}(i,j),$$
where $x_{c}$ indicates the input features at c-th channel. Moreover, the global spatial information in the 3 × 3 branch is encoded using 2D global average pooling. Subsequently, the second spatial attention map, which maintains accurate spatial positional information, is generated. Each group’s final output feature map is obtained by aggregating the two spatial attention weights and applying a Sigmoid function. This method effectively captures pairwise relationships at the pixel level and emphasizes the global context across all pixels. The final output of the EMA module matches the dimensions of X, which ensures efficiency and compatibility with modern architectures.

## 3. Experiments and Discussion

### 3.1. Environment and Configuration

In this study, we trained the ship detection models on a server in an Ubuntu 22.04 Linux environment with an Intel^®^ Core™ I9-10850K CPU (Santa Clara, CA, USA) and an NVIDIA RTX 3080 GPU (Santa Clara, CA, USA). The deep learning framework was Pytorch.

Consistent hyperparameters were utilized throughout the training phases for all experiments. The specific hyperparameters of the model used during the training are detailed in Table 1.

For the parameter settings of the Stable Diffusion model, we used the stable-diffusion-v1-5 version. The sampling steps *T* were set to 50, and the denoising strength was set to 0.35. Through the data augmentation method we proposed, we obtained an augmented dataset. Figure 8 shows some example images of the augmented dataset. 

### 3.2. Performance Evaluation of the Model

In this study, we evaluated the model’s performance by considering both detection performance and computational efficiency. Average precision (AP) and mean average precision (mAP) are commonly used metrics for object detection. AP is obtained from the precision-recall curve, calculated at various confidence thresholds, and integrated over the curve. The equations for precision, recall, and AP are as follows:(5)$$P=\frac{TP}{TP+FP},$$
(6)$$R=\frac{TP}{TP+FN}$$
(7)$$AP=\sum_{k}\left(P(k)\times ∆R(k)\right),$$

True positive (TP) refers to the number of samples correctly identified as positive by the model. False positive (FP) refers to the number of samples incorrectly identified as positive when they are negative. False negative (FN) represents the number of samples incorrectly identified as negative when they are positive. P(k) represents precision in the k-th samples, and ∆R(k) refers to the difference between the recall at position k and the recall at position k−1.

Meanwhile, mAP represents the average AP across multiple categories, comprehensively evaluating the model’s performance across diverse objects. The equation for mAP is as follows:(8)$$mAP=\frac{1}{k}\sum_{1}^{k}AP_{i},$$
where k refers to the number of object classes. Unless otherwise stated, this paper’s AP, mAP, and related metrics were obtained through the test set.

Floating-Point Operations (FLOPs), a measure of the total number of floating-point operations required by the neural network model, are used to evaluate the computational demand of the model. A higher FLOPs count indicates more significant computational requirements. The number of model parameters is the total number of tunable parameters to be learned or optimized in the neural network model. A smaller number of model parameters reduces the storage and computational resource requirements.

### 3.3. Data Augmentation Method Performance

To evaluate the impact of data augmentation methods, we trained the original dataset using three methods: traditional data augmentation (rotation, flip, contrast adjustment, etc.), stable diffusion, and our proposed method. All applied to the original YOLOv8n OBB model. In this study, all experiments are conducted using OBB models. To simplify model names, the term “OBB” is omitted from model names in the following sections. To avoid the interference caused by the image pixel, all original and generated images were uniformly resized to 640 × 640 pixels using the letterbox method. To avoid interference caused by data volume, all three data augmentation methods were used to increase the training set to approximately 4500 images, while the validation and test sets remained unchanged. In addition, we adjusted the number of generated images for different classes to expand the hard-to-sample ship instances and increase the number of samples for the underrepresented classes in the dataset. Table 2 shows the number of ship instances in different training sets.

The data augmentation results for the original dataset, traditional data augmentation, stable diffusion, and our proposed method are presented in Table 3. AP (@0.5) refers to the average precision calculated with an IoU threshold of 0.5. Specifically, it measures the precision and recall of the model when the predicted bounding box overlaps with the truth box by at least 50%. mAP (@0.5) refers to the mean average precision calculated using an IoU threshold of 0.5. mAP (@0.5:0.95) refers to the mean average precision calculated using an IoU threshold ranging from 0.5 to 0.95.

From the results, our proposed method showed an overall improvement of 2.8% in the mAP50 metric compared to the method using the original dataset. Our proposed method achieves an overall improvement of 0.9% compared to the traditional method. Specifically, the AP (@0.5) metrics for the ship classes “Oil Tanker” and “Others” showed a noticeable improvement. And the AP (@0.5) metrics for the ship classes “Bulk Carrier”, “Container Ship” and “Passenger Ship” did not show a noticeable improvement. After analysis, we found that the “Oil Tanker” and “Others” ship classes had fewer original instances and exhibited a variety of models and complex features. The ‘Bulk Carrier’ and ‘Container Ship’ categories had sufficient original instances for the model to learn enough features. The “Passenger Ship” had fewer original instances. However, it had only two types of models during the sampling period in the Pearl River waterway, and the features of the “Passenger Ship” were not complex. It indicated that our proposed method exhibited strong performance in scenarios with limited original data and complex object features. Our proposed data augmentation method can randomly generate images with new features based on the original images, increasing the number of low-volume ship instances and thereby improving the detection rate of the trained model for the low-volume ship instances.

### 3.4. Performance of Neck Structures

Inspired by the structure of BiFPN, we used the feature map P2 as input and improved the structure of the YOLOv8 neck to enhance the feature fusion capability and improve the ship detection performance. As shown in Table 4, the models were all trained on the datasets obtained using our proposed method. Compared to the original YOLOv8n model, the YOLOv8n-BiFPN model increased the mAP (@0.5) and mAP (@0.5:0.95) by 0.5% and 0.4%, respectively. The number of model parameters decreased by 0.1 million, while the FLOPs increased by 9.5 billion. Additionally, the inference time per image increased by 1.7 ms.

### 3.5. Performance of Attention Modules

Besides the EMA module, CBAM and CA modules are frequently applied in improved object detection models. To evaluate the effectiveness of the EMA module, we added the CBAM and CA modules at the same positions in the YOLOv8n-BiFPN model for comparison. As shown in Table 5, the experimental results of the EMA module are the best, with mAP (@0.5) being 1.9% higher than the CBAM and 3.7% higher than the CA, while the inference time per image is approximately the same.

### 3.6. Overall Performance Evaluation

To evaluate the overall performance of our proposed ship detection model, we conducted comparative experiments involving several models: the YOLOv8n model trained on the original dataset, the YOLOv8n model trained on our augmented dataset, our proposed YOLOv8n-BiFPN-EMA model trained on the augmented dataset, and other state-of-the-art detection models trained on the augmented dataset.

Table 6 presents the validation results of the models on the test set. In summary, the YOLOv8n-BiFPN-EMA model we proposed performed the best. Our proposed model achieved the highest mAP (@0.5) and mAP (@0.5:0.95) alongside reduced network parameters. Specifically, our proposed model outperformed Faster R-CNN and S2ANet with mAP (@0.5) improvements of 1.7% and 8%, significantly reducing the inference time per image. Although the YOLOv5n model demonstrated the shortest inference time, its mAP (@0.5) was limited to 80.7%. Compared to the YOLOv7 model, our model achieved a higher mAP (@0.5) and reduced the inference time. These results indicated that our proposed model struck an optimal balance between accuracy and detection speed, making it particularly well-suited for real-time ship detection using camera-equipped UAVs. Figure 9 compares our proposed YOLOv8n-BiFPN-EMA model more intuitively with other models.

To visually assess the impact of our proposed data augmentation method and the performance of the proposed model, we employed three models: the YOLOv8n model trained on the original dataset, the YOLOv8n model trained on the augmented dataset, and our proposed YOLOv8n-BiFPN-EMA model trained on the augmented dataset. These models were used to generate predictions on a subset of test images.

Figure 10, Figure 11 and Figure 12, respectively illustrated the visual ship detection results of the three models. From the three figures, it can be seen that when facing multi-viewpoint and multi-scale ship detection tasks, the YOLOv8n-BiFPN-EMA model identified more ship instances, had a higher confidence threshold for ship detections, and exhibited fewer missed and false detections. It indicates that our improvements to the YOLOv8n model were effective. The neck structure, improved by incorporating the BiFPN structure, integrating the feature map P2, and adding the EMA attention module, effectively enhanced the model’s capability to detect multi-viewpoint and multi-scale ship instances. Additionally, we could visually observe that the YOLOv8n model with the original dataset had a significantly lower detection rate and confidence threshold for the “Oil Tankers” and “Others” ship classes compared to the other two models. It further confirmed our conclusion in Section 3.3. Based on stable diffusion, our proposed data augmentation method performed well when faced with limited original data and complex object features.

## 4. Conclusions

This study aims to enhance real-time ship detection capability, making ship management and monitoring in the Pearl River waterway more efficient. To this end, we propose a data augmentation method based on stable diffusion to improve the models’ detection efficiency and address the data scarcity issue in this study. An improved ship detection model based on the YOLOv8n OBB model is proposed, with enhancements to the neck structure through the BiFPN structure and EMA module. Among these optimizations, the data augmentation method aims to improve the quality and quantity of the dataset, achieving better training results despite the small number of datasets. Improving the ship detection model structure is intended to enhance detection performance and precision while reducing the number of parameters for easy deployment. Experimental results validate the effectiveness of these optimization methods, achieving an mAP (@0.5) of 92.3%, an mAP (@0.5:0.95) of 77.5%, a reduction of 0.8 million in model parameters, and a detection speed that satisfies real-time ship detection requirements.

In the future, we plan to deploy our proposed real-time ship detection model on a high-performance ground computing platform through an efficient image transmission network. This approach will address UAV platforms’ low computing power issues and enable efficient ship detection. In addition, we will continue to explore the application of our proposed data augmentation method based on stable diffusion to other tasks.

## Figures and Tables

**Figure 1 sensors-24-05850-f001:**

A horizontal bounding box and an oriented bounding box enclose the same ship instance. (**a**) Original image; (**b**) Horizontal bounding box; (**c**) Oriented bounding box.

**Figure 2 sensors-24-05850-f002:**
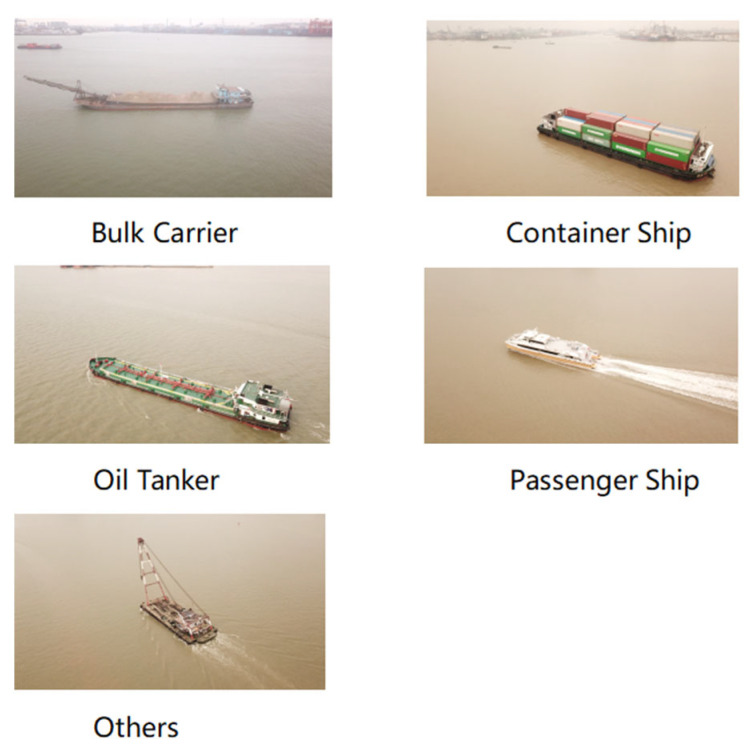
Each of the five classes of ship instances.

**Figure 3 sensors-24-05850-f003:**
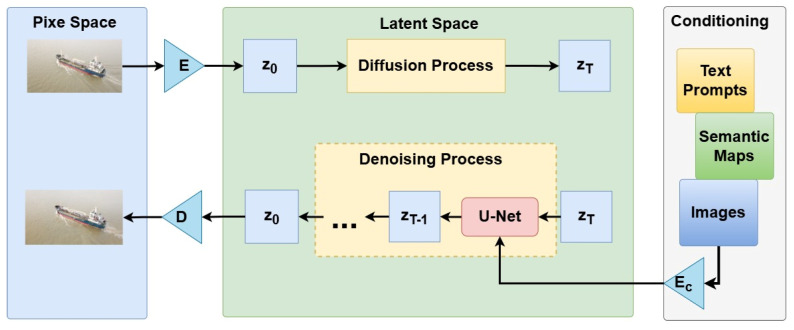
The structure of stable diffusion model.

**Figure 4 sensors-24-05850-f004:**
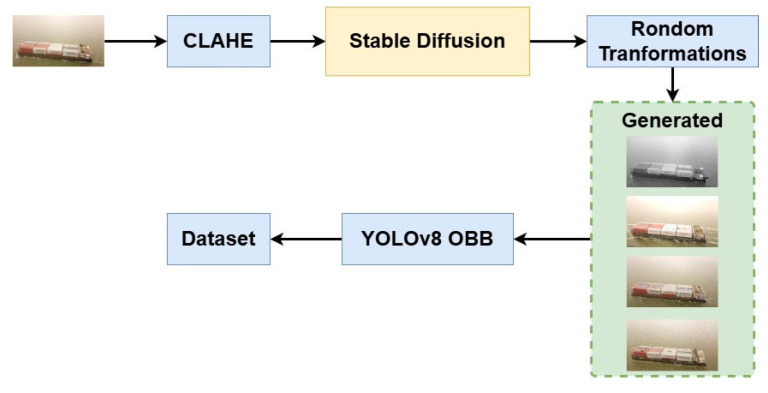
The improved data augmentation method.

**Figure 5 sensors-24-05850-f005:**
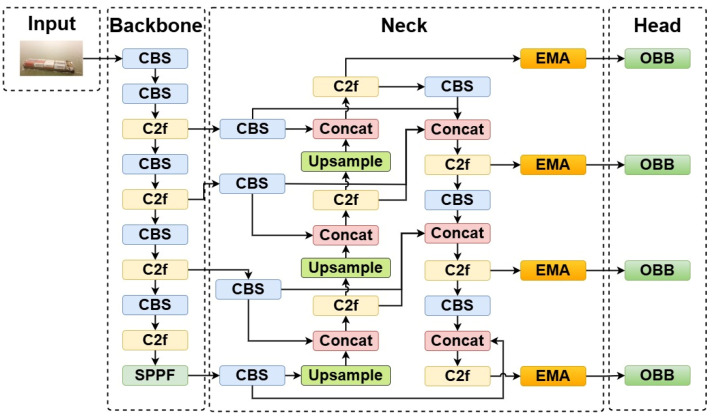
The architecture of the improved YOLOv8 OBB model.

**Figure 6 sensors-24-05850-f006:**
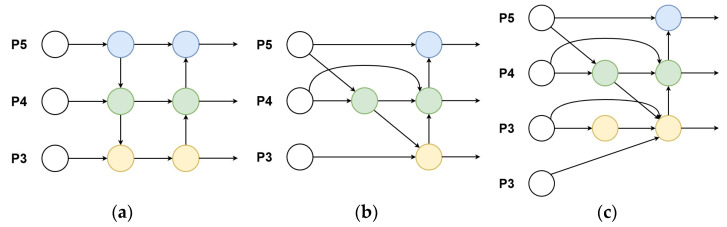
The structures of PANet, BiFPN, and improved BiFPN. (**a**) PANet; (**b**) BiFPN; (**c**) Improved BiFPN. Circles of different colors represent different feature maps.

**Figure 7 sensors-24-05850-f007:**
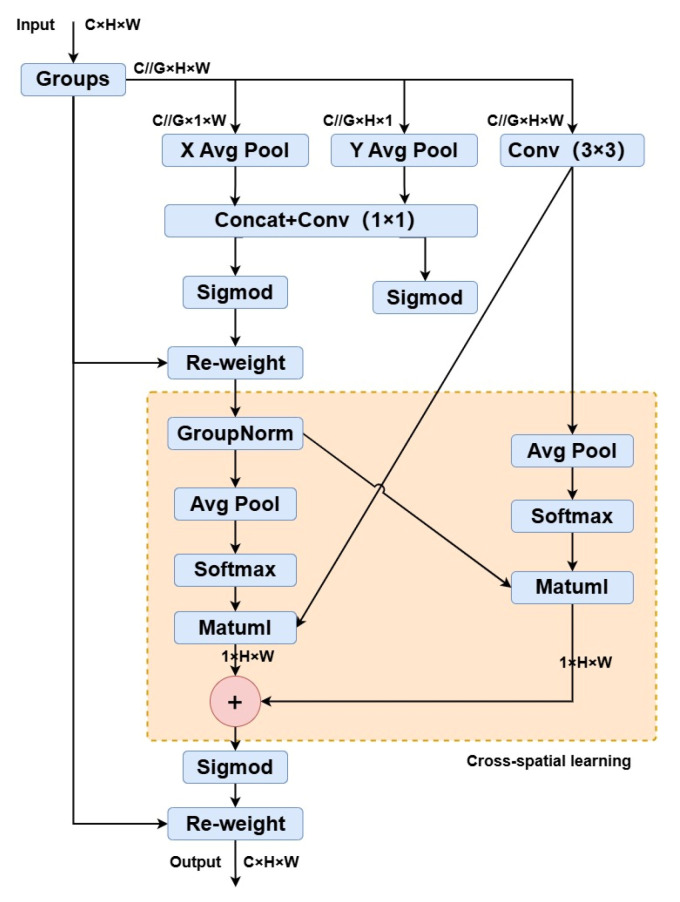
The structure of the EMA module.

**Figure 8 sensors-24-05850-f008:**
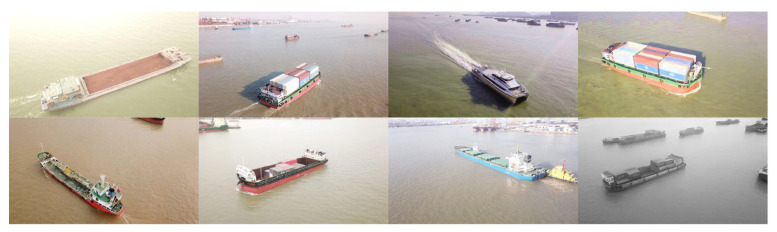
Some example images of the augmented dataset.

**Figure 9 sensors-24-05850-f009:**
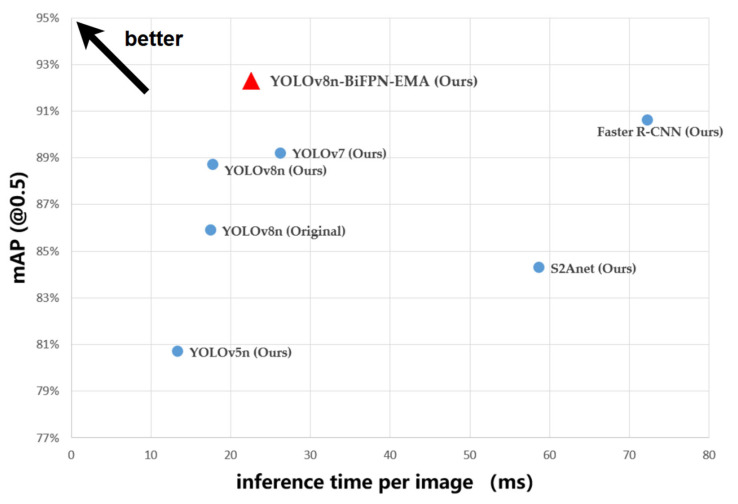
Comparison of the models.

**Figure 10 sensors-24-05850-f010:**
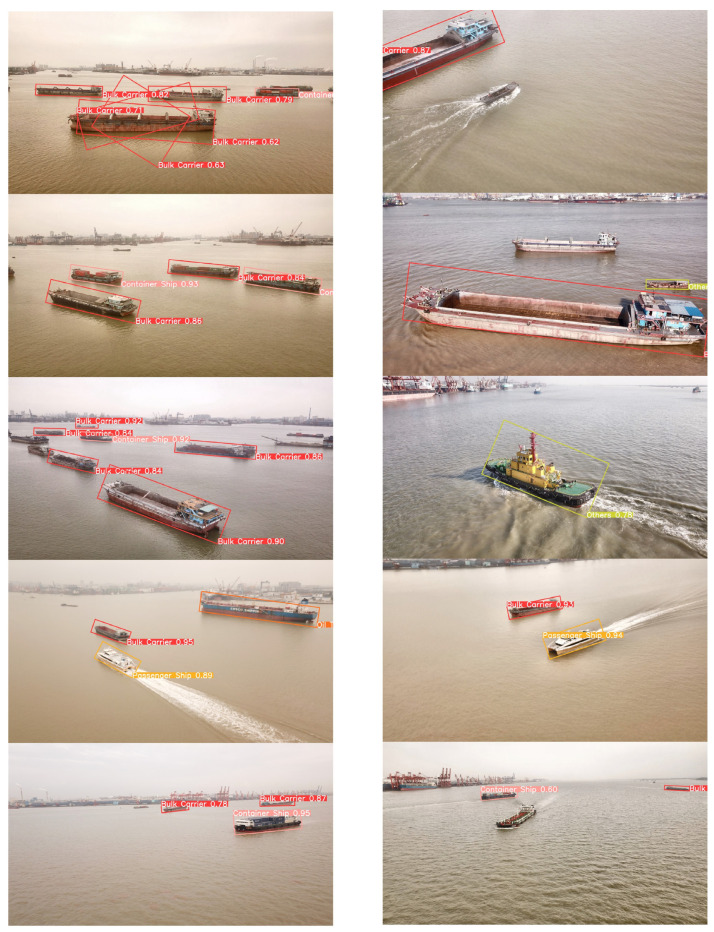
YOLOv8n (Original).

**Figure 11 sensors-24-05850-f011:**
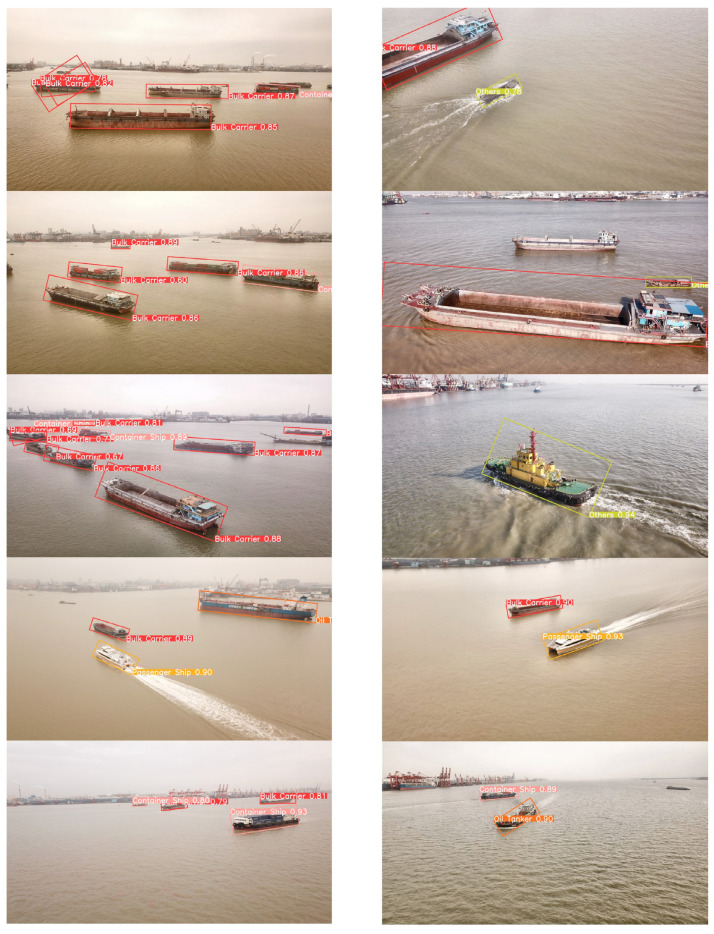
YOLOv8n (Ours).

**Figure 12 sensors-24-05850-f012:**
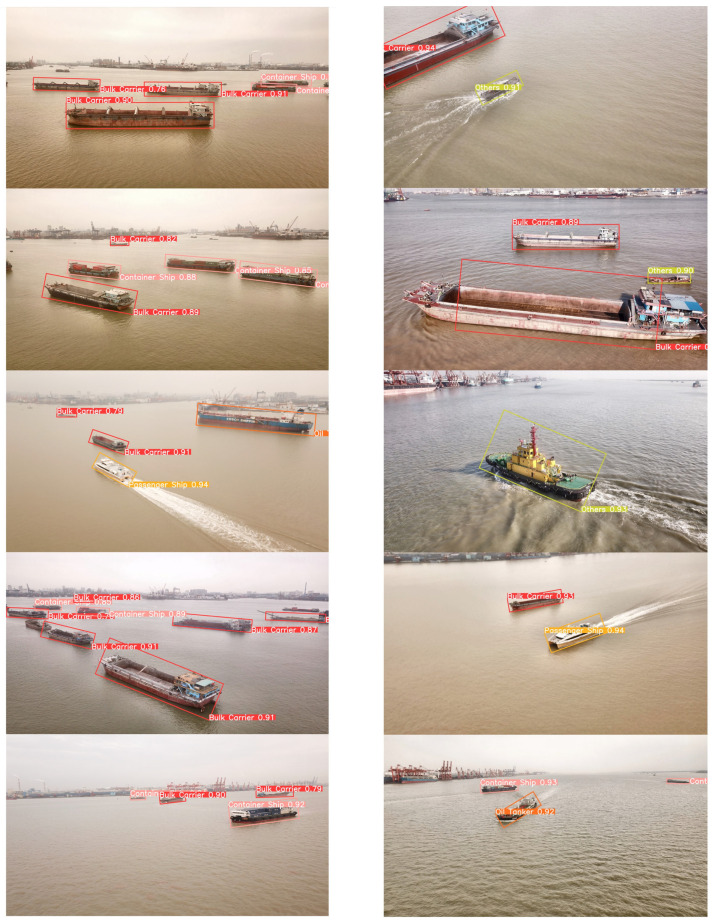
YOLOv8n-BiFPN-EMA (Ours).

**Table 1 sensors-24-05850-t001:** The hyperparameters of the model.

Hyperparameters	Value
Learning Rate	0.01
Image Size	640 × 640
Epochs	100
Optimizer	SGD
Weight Decay	0.0005
Momentum	0.937

**Table 2 sensors-24-05850-t002:** The number of ship instances.

Method	Images	Bulk Carrier	Container Ship	Oil Tanker	Passenger Ship	Others
Original	441	410	214	85	19	57
Traditional	4396	4077	2123	849	303	456
Stable diffusion	4649	3896	2448	1113	399	969
Ours	4649	3869	2448	1113	399	969

**Table 3 sensors-24-05850-t003:** The data augmentation results.

Method	AP(@0.5)					mAP (@0.5)	mAP (@0.5:0.95)
	Bulk Carrier	Container Ship	Oil Tanker	Passenger Ship	Others		
Original	84.8%	83.2%	90.2%	96.5%	75.1%	85.9%	71.8%
Traditional	85.3%	86.2%	94.7%	94.4%	78.2%	87.8%	74.3%
Stable diffusion	81.1%	84.6%	98.8%	93.8%	78.4%	87.3%	73.6%
Ours	84.4%	83.7%	99.5%	94.7%	81.1%	88.7%	75.0%

**Table 4 sensors-24-05850-t004:** The performance of neck structures.

Model	mAP (@0.5)	mAP (@0.5:0.95)	Parameters (M)	FLOPs (B)	ms/img
YOLOv8n (Ours)	88.7%	75.0%	3.1	8.4	17.8
YOLOv8n- BiFPN (Ours)	89.2%	75.4%	2.3	17.9	19.5

**Table 5 sensors-24-05850-t005:** The performance of attention modules.

Model	mAP (@0.5)	mAP (@0.5:0.95)	Parameters (M)	FLOPs (B)	ms/img
YOLOv8n- BiFPN (Ours)	89.2%	75.4%	2.3	17.9	19.5
+CA	88.6%	73.6%	2.3	18.2	21.8
+CBAM	90.4%	74.9%	2.3	18.2	22.1
+EMA	92.3%	77.5%	2.3	18.3	22.6

**Table 6 sensors-24-05850-t006:** The overall performance.

Model	mAP (@0.5)	mAP (@0.5:0.95)	Parameters (M)	FLOPs (B)	ms/img
YOLOv8n (Original)	85.9%	71.8%	3.1	8.4	17.5
YOLOv8n	88.7%	75.0%	3.1	8.4	17.8
(Ours)					
YOLOv8n-BiFPN	92.3%	77.5%	2.3	18.3	22.6
-EMA (Ours)					
Faster R-CNN	90.6%	74.3%	41.8	207.8	72.3
(Ours)					
S2ANet (Ours)	84.3%	69.2%	34.2	160.1	58.7
YOLOv5n (Ours)	80.7%	63.8%	2.1	7.3	13.4
YOLOv7 (Ours)	89.2%	74.6%	37.4	105.4	26.3

## Data Availability

Data are contained within the article.

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
