# Peer review of "An Improved YOLOv8 OBB Model for Ship Detection through Stable Diffusion Data Augmentation"

_sensors, 2024, doi:10.3390/s24175850_

Round 1

Reviewer 1 Report

Comments and Suggestions for Authors

Dear editor:

Thank you for inviting me to evaluate the paper titled " An Improved YOLOv8 OBB Model for Ship Detection Through Stable Diffusion Data Augmentation . In this paper, the authors investigated the problem of real-time ship detection by UAVs. The authors applied data augmentation based on stable diffusion and made improvements to the YOLOv8 model, which contribute to the innovation of this paper. On the whole, this paper is of good quality. After addressing some minor issues, I recommend that this paper can be accepted. These following are some suggestions for this manuscript:

1.In Abstract and Conclusion, ' mAP@50' and ' mAP (@0.5)' are used interchangeably in the paper. I recommend standardizing them to ' mAP (@0.5)' throughout.

2. In the Data capture section, I suggest providing a more detailed description of the process of capturing image data.

3. To better illustrate the effects of the data augmentation method in this paper, I suggest presenting some example images from the augmented dataset.

Comments on the Quality of English Language

Very Good

Author Response

Comments 1:In Abstract and Conclusion, ' mAP@50' and ' mAP (@0.5)' are used interchangeably in the paper. I recommend standardizing them to ' mAP (@0.5)' throughout.

Response 1:Thank you for pointing this out. We have modified the mistake in  Abstract and Conclusion.

Comments 2:In the Data capture section, I suggest providing a more detailed description of the process of capturing image data.

Response 2:Thank you for your suggestion. We have changed the description of the process of capturing image data.

Comments 3:To better illustrate the effects of the data augmentation method in this paper, I suggest presenting some example images from the augmented dataset.

Response 3:We agree with this comment. We have presented some example images from the augmented dataset  in figure 8..

Reviewer 2 Report

Comments and Suggestions for Authors

1. I don't really see your innovativeness from the article out, it seems to be just a blend of other people's innovativeness and adding a testing head to the structure.

2. the experimental data is seriously insufficient, it should be compared with other mainstream models, not just with YOLOv8.

3. Experiments should be conducted on public datasets to test performance, citing other people's experimental results for comparison to prove the validity of your model.

4. In the beginning, there is a description of the model base used is YOLOv8-OBB, but in the experimental part of the use of YOLOv8, whether it needs to be modified.

Author Response

Thank you for your thoughtful review of this manuscript. We have carefully considered your feedback and have made revisions accordingly. Below, you will find responses to each of your comments.

Comments 1: I don't really see your innovativeness from the article out, it seems to be just a blend of other people's innovativeness and adding a testing head to the structure.

Response 1: Thank you for pointing this out. In this study, we proposed a data augmentation method based on stable diffusion for expanding the dataset. And we improved the YOLOv8n OBB model by incorporating the BiFPN structure and EMA module. We applied our proposed data augmentation method and our proposed model to ship detection using camera-equipped UAVs.

Comments 2: The experimental data is seriously insufficient, it should be compared with other mainstream models, not just with YOLOv8.

Response 2: Thank you for pointing this out very much. We agree with this comment. We have added four groups of experiments with other state-of-the-art detection models in Section 3.6.  Furthermore, we have visualized the comparative results in Figure 9 to facilitate a more intuitive understanding of the differences in model performance.

Comments 3: Experiments should be conducted on public datasets to test performance, citing other people's experimental results for comparison to prove the validity of your model.

Response 3: Thank you for your feedback regarding the experimental dataset. We agree that evaluating models on public datasets facilitates comparisons with results from other models. And experiments on custom-built datasets also offer benefits. Our experimental dataset is a custom-built dataset augmented using our proposed method. The reasons for this approach are as follows: 1. There is a lack of  public datasets that meet our requirements. In our study,datasets for Oriented Bounding Box (OBB) detection models differ from those used in conventional object detection tasks, as they require annotations with Oriented Bounding Boxes. 2. A custom-built dataset improves model performance in real-world applications. Ships exhibit varying feature depending on their location. Therefore, ship images collected from the actual waterway are most representative of the application scenario for our task.

Comments 4: In the beginning, there is a description of the model base used is YOLOv8-OBB, but in the experimental part of the use of YOLOv8, whether it needs to be modified.

Response 4: Thank you for pointing out the error. All the models we used are Oriented Bounding Box (OBB) detection models. To simplify model names, we omit the term “OBB”. Following your feedback,we have added the relevant explanation in line 357. 

Thank you once again for your valuable feedback on this manuscript. Your comments have been instrumental in improving the quality of the manuscript. We appreciate your time and effort in reviewing our submission.

Reviewer 3 Report

Comments and Suggestions for Authors

This paper tries to tackle a pertinent problem but falls short in establishing itself as an effective alternative. While extensive ablation studies have proven the effectiveness of each component, they lack overall comparisons. Furthermore, the literature review does not cite multiple other studies on utilizing drones for ship detection. These are some of the ones I found with a google search:
1. Wang, Quanzheng, et al. "A YOLOv7-Based Method for Ship Detection in Videos of Drones." Journal of Marine Science and Engineering 12.7 (2024): 1180.
2. Cheng, Shuxiao, Yishuang Zhu, and Shaohua Wu. "Deep learning based efficient ship detection from drone-captured images for maritime surveillance." Ocean Engineering 285 (2023): 115440.
3. Li, YongShuai, et al. "Maritime vessel detection and tracking under UAV vision." 2022 International Conference on Artificial Intelligence and Computer Information Technology (AICIT). IEEE, 2022.

Multiple other recent publications have been published in this field, and you need to improve your literature review by adding these studies as well. While you go into details about object detection, you lack a detailed review of the drone-based ship detection method, which is the main topic of your paper. 

Similarly, you only compare to YOLOv8n for overall comparison. This fails to establish your method as superior in comparison. There are various other variations of YOLOv8 that you could compare your method against and show the results to prove the superiority of your method.

Comments on the Quality of English Language

There are multiple grammatical errors in the paper which need to be fixed. In Line 70, you have "side view, rear view and et al [17]." This is not the correct usage of et al. Similarly, in line 72, "instances lead to ship instances have multiple scales". This should have been "instances lead to ship instances having multiple scales". 

Like this, there are multiple grammatical errors throughout the paper. You will have to conduct a thorough check to fix the grammar. 

Author Response

Thank you for your thoughtful review of this manuscript. We have carefully considered your feedback and have made revisions accordingly. Below, you will find responses to each of your comments.

Comments 1:Multiple other recent publications have been published in this field, and you need to improve your literature review by adding these studies as well. While you go into details about object detection, you lack a detailed review of the drone-based ship detection method, which is the main topic of your paper. 

Response 1: Thank you for pointing this out very much. We agree with this comment. And we have added a detailed review of the drone-based ship detection method in lines 55 to 68. 

Comments 2:Similarly, you only compare to YOLOv8n for overall comparison. This fails to establish your method as superior in comparison. There are various other variations of YOLOv8 that you could compare your method against and show the results to prove the superiority of your method.

Response 2: We agree with this comment. We have added four groups of experiments with other state-of-the-art detection models in Section 3.6.  Furthermore, we have visualized the comparative results in Figure 9 to facilitate a more intuitive understanding of the differences in model performance.

Comments 3:There are multiple grammatical errors in the paper which need to be fixed. In Line 70, you have "side view, rear view and et al [17]." This is not the correct usage of et al. Similarly, in line 72, "instances lead to ship instances have multiple scales". This should have been "instances lead to ship instances having multiple scales". 

Response 3: Thank you for pointing out the errors in our English writing. We have corrected the errors you pointed out. Additionally, we have checked and revised the English grammar throughout the manuscript.

Thank you once again for your valuable feedback on this manuscript. Your comments have been instrumental in improving the quality of the manuscript. We appreciate your time and effort in reviewing our submission.

Round 2

Reviewer 2 Report

Comments and Suggestions for Authors

1.This manuscript lacks generalization comparison experiments of the proposed model (such as experiments on datasets in similar fields), and it is recommended to add them.

Comments on the Quality of English Language

There are still some grammatical errors in the manuscript that need further checking and improvement.

Author Response

Thank you for your thorough review of our manuscript. We have carefully considered your feedback and have revised the manuscript accordingly. Below, we provide detailed responses to each of your comments.

Comments 1:This manuscript lacks generalization comparison experiments of the proposed model (such as experiments on datasets in similar fields), and it is recommended to add them.

Response 1: Thank you very much for your recommendation. In our study, we employed Oriented Bounding Box (OBB) detection models for ship detection, necessitating the use of OBB-specific datasets. The primary public OBB ship detection datasets, DOTA and HRSC2016, consist of high-altitude satellite images captured from a single top-down viewpoint at long distances. These datasets are unsuitable for our research, which focuses on low-altitude, multi-viewpoint ship images. The primary public OBB ship detection datasets, DOTA and HRSC2016, consist of high-altitude satellite images captured from a single top-down viewpoint at long distances. These datasets are unsuitable for our research, which focuses on low-altitude, multi-viewpoint ship images. While conventional public ship detection datasets annotated with Horizontal Bounding Boxes (HBB) exist, they would need to be re-annotated with OBB labels to be applicable to our task. This re-annotation process alters the original dataset content, diminishing its reference value. Consequently, we opted to create a custom ship detection dataset tailored to the actual conditions of our task, thereby better meeting our requirements and enhancing ship detection performance. The improvements in our proposed model are derived from the training results on this custom ship detection dataset. Experimental results on datasets in similar fields are less relevant as references. Additionally, after addressing data sensitivity concerns and obtaining the necessary permissions, we are considering open-sourcing our ship detection dataset to make it publicly available.

Comments 2:There are still some grammatical errors in the manuscript that need further checking and improvement.

Response 2:Thank you for pointing out the grammatical errors in our manuscript. We have thoroughly rechecked and revised the English grammar throughout the entire manuscript.

Thank you once again for your valuable recommendations. Your comments have been instrumental in enhancing the quality of the manuscript. We sincerely appreciate the time and effort you have invested in reviewing our submission.